# Distribution and Ecology of Decapod Crustaceans in Mediterranean Marine Caves: A Review

**Carlo Nike Bianchi** [1,*], **Vasilis Gerovasileiou** [2,3], **Carla Morri** [1] **and Carlo Froglia** [4]

1. Department of Earth, Environmental and Life Sciences, University of Genoa, 16132 Genova, Italy; carla.morri.ge@gmail.com
2. Department of Environment, Faculty of Environment, Ionian University, 29100 Zakynthos, Greece; vgerovas@ionio.gr
3. Hellenic Centre for Marine Research (HCMR), Institute of Marine Biology, Biotechnology and Aquaculture (IMBBC), 71500 Heraklion, Crete, Greece; vgerovas@hcmr.gr
4. CNR-IRBIM, National Research Council, Institute for Marine Biological Resources and Biotechnologies, 60125 Ancona, Italy; c.froglia@alice.it
* Correspondence: carlonike.bianchi.ge@gmail.com

**Abstract:** Decapod crustaceans are important components of the fauna of marine caves worldwide, yet information on their ecology is still scarce. Mediterranean marine caves are perhaps the best known of the world and may offer paradigms to the students of marine cave decapods from other geographic regions. This review summarizes and updates the existing knowledge about the decapod fauna of Mediterranean marine caves on the basis of a dataset of 76 species from 133 caves in 13 Mediterranean countries. Most species were found occasionally, while 15 species were comparatively frequent (found in at least seven caves). They comprise cryptobiotic and bathyphilic species that only secondarily colonize caves (secondary stygobiosis). Little is known about the population biology of cave decapods, and quantitative data are virtually lacking. The knowledge on Mediterranean marine cave decapods is far from being complete. Future research should focus on filling regional gaps and on the decapod ecological role: getting out at night to feed and resting in caves during daytime, decapods may import organic matter to the cave ecosystem. Some decapod species occurring in caves are protected by law. Ecological interest and the need for conservation initiatives combine to claim for intensifying research on the decapod fauna of the Mediterranean Sea caves.

**Keywords:** species inventory; zoogeography; ecology; species richness; depth preference; cave zonation; secondary stygobiosis; trophic depletion; protected species

## 1. Introduction

Members of order Decapoda are among the crustaceans most familiar to the general public and include species of commercial interest for fisheries, such as lobsters, shrimps and prawns [1]. They total over 14,000 extant species [2], which have colonized virtually all aquatic habitats, with a few species even thriving on land. Crustacean decapods are one of the most important groups—in terms of both species richness and ecological roles [3]—in the marine realm, where they occur from the upper intertidal zone to hadal depths [4], with several species adapted to extreme environments, such as hydrothermal vents and cold seeps [5].

Marine caves are another example of an extreme environment [6] that hosts a rich decapod fauna [7–9]. Rocky reefs and coral reefs around the world harbour cavities of various size at or below sea level and, therefore, filled with marine water. Several decapod taxa new to science have been described from, and occur only in, marine caves. Yet, information on the distribution and ecology of cave decapods is still largely fragmentary.

The rocky coasts of the Mediterranean Sea are particularly rich in marine caves [10], whose geology, biology and ecology have been studied with continuity for decades [11–13]. Thus, Mediterranean marine caves are perhaps the best known of the world ocean and may offer paradigms and theories to the students of marine cave decapods from other geographic regions.

This review summarizes and updates the existing knowledge about the decapod fauna of Mediterranean marine caves on the basis of a dataset assembled within the frame of a broader initiative aiming at cataloguing and assessing the biodiversity of Mediterranean marine caves [12,14]. The database incorporates records from either scientific publications or grey literature; unpublished observations by the authors in marine caves of Greece and Italy were purposely added for this review. The final decapod checklist will be mobilised to the World Register of Marine Cave Species (WoRCS, http://www.marinespecies.org/worcs/), a thematic species database of the World Register of Marine Species (WoRMS, http://www.marinespecies.org, accessed on 23 February 2022), which aims at creating a comprehensive taxonomic and ecological database of species known from marine caves worldwide [15]. When available, ecological and spatial information on the caves was also considered [14]. A thorough ecological analysis has been possible for the Grotta Marina di Bergeggi (NW Italy), which has been extensively studied since the 1970s [16–18] and where the distribution of decapod species within the cave can be compared with the available topographic information and environmental parameters [19,20]. Anchialine caves, which contain water bodies of marine origin but have reduced or no connection with the sea [21] and harbour a distinctive fauna [22], are not covered by this review: Mediterranean anchialine caves are still too little studied for decapods [23].

## 2. Species Inventory and Frequency

Only 68 out of the 370 literature sources examined (18%) provided information on decapod occurrence. All the decapod species found in at least one Mediterranean marine cave were considered. In total, 76 decapod species have been reported from 133 caves (or cave groups) in 13 countries (Figure 1). Most of these caves (52) are located in Italy, followed by Greece (23 caves), Montenegro (13) and France (11). Spain and Turkey provided nine caves each, Croatia seven and Albania four, while Cyprus, Israel, Lebanon, Malta and Morocco provided only one cave (or cave group) each.

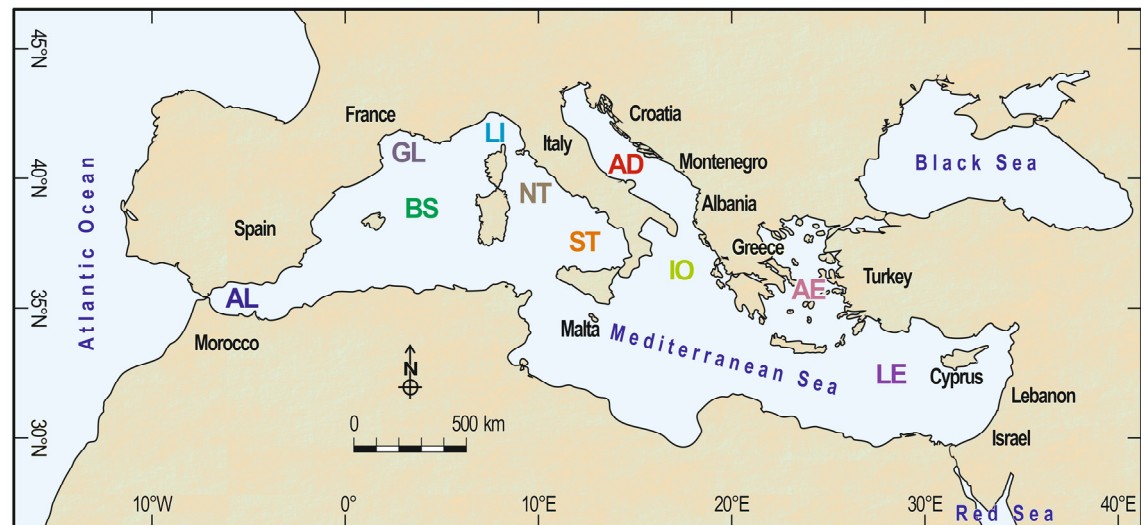

**Figure 1.** Map of the Mediterranean Sea with countries and marine sectors where cave decapods have been reported. AL = Alboran Sea; BS = Balearic and Sardinia seas; GL = Gulf of Lion; LI = Ligurian Sea; NT = North Tyrrhenian Sea; ST = South Tyrrhenian Sea; AD = Adriatic Sea; IO = Ionian Sea; AE = Aegean Sea; LE = Levantine Sea.

The present total of 76 species (Table 1) represents a momentous increase with respect to the earliest accounts of Riedl [7], who listed 11 decapod species from Mediterranean marine caves, and Parenzan [24], who listed seven species from Italian marine caves. The first study entirely devoted to the decapod fauna in Mediterranean marine caves was that of Gili and Macpherson [25], who collected 11 species in two caves of Mallorca (Balearic Islands, Spain). Pessani and Manconi [26] found 15 species in five marine caves of the Sorrentine Peninsula (Italy), while Pipitone and Vaccaro [27] recorded eight species in four marine caves of the Island of Ustica (Sicily, Italy). The synthesis by Manconi and Pessani [28] reported 25 species from 21 Italian marine caves. Denitto et al. [29,30] studied the decapod fauna (14 species) of a marine cave in SE Italy, focussing on *Palaemon* spp. and the west-African species *Herbstia nitida*, recorded for the first time in the Mediterranean Sea. In semi-submerged marine caves of Montenegro, Mačić et al. [31] found five species. Two species new to science were described from Mediterranean marine caves: *Odontozona addaia*, from Menorca (Balearic Islands, Spain) [32], and *Salmoneus sketi*, from Lavernaka Island (Croatia) [33].

**Table 1.** List of the species of decapod crustaceans found in marine caves of the Mediterranean Sea.

| Infraorder | Family | Genus and species |
|---|---|---|
| Stenopodidea | Stenopodidae | *Odontozona addaia* Pretus, 1990 |
| | | *Stenopus spinosus* Risso, 1827 |
| Caridea | Palaemonidae | *Balssia gasti* (Balss, 1921) |
| | | *Brachycarpus biunguiculatus* (Lucas, 1846) |
| | | *Gnathophyllum elegans* (Risso, 1816) |
| | | *Palaemon adspersus* Rathke, 1837 |
| | | *Palaemon elegans* Rathke, 1837 |
| | | *Palaemon serratus* (Pennant, 1777) |
| | | *Palaemon xiphias* Risso, 1816 |
| | | *Periclimenes amethysteus* (Risso, 1827) |
| | | *Periclimenes scriptus* (Risso, 1822) |
| | | *Urocaridella pulchella* Yokes & Galil, 2006 |
| | Alpheidae | *Alpheus dentipes* Guérin, 1832 |
| | | *Athanas nitescens* (Leach, 1814) |
| | | *Salmoneus sketi* Fransen, 1991 |
| | | *Synalpheus gambarelloides* (Nardo, 1847) |
| | Hippolytidae | *Caridion* sp. |
| | | *Hippolyte holthuisi* Zariquiey Alvarez, 1953 |
| | | *Saron marmoratus* (Olivier, 1811) |
| | Lysmatidae | *Lysmata nilita* Dohrn & Holthuis, 1950 |
| | | *Lysmata seticaudata* (Risso, 1816) |
| | Pandalidae | *Plesionika narval* (Fabricius, 1787) |
| | Thoridae | *Eualus cranchii* (Leach, 1817) |
| | | *Eualus occultus* (Lebour, 1936) |
| | | *Eualus sollaudi* (Zariquiey Cenarro, 1936) |
| Astacidea | Nephropidae | *Homarus gammarus* (Linnaeus, 1758) |
| Achelata | Palinuridae | *Palinurus elephas* (Fabricius, 1787) |
| | Scyllaridae | *Scyllarides latus* (Latreille, 1803) |
| | | *Scyllarus arctus* (Linnaeus, 1758) |
| | | *Scyllarus pygmaeus* (Bate, 1888) |
| Anomura | Galatheidae | *Galathea dispersa* Bate, 1859 |
| | | *Galathea intermedia* Lilljeborg, 1851 |
| | | *Galathea nexa* Embleton, 1836 |

| | | *Galathea strigosa* (Linnaeus, 1761) |
|---|---|---|
| | Munididae | *Munida rugosa* (Fabricius, 1775) |
| | Porcellanidae | *Pisidia bluteli* (Risso, 1816) |
| | | *Porcellana platycheles* (Pennant, 1777) |
| | Diogenidae | *Calcinus tubularis* (Linnaeus, 1767) |
| | | *Clibanarius erythropus* (Latreille, 1818) |
| | | *Dardanus arrosor* (Herbst, 1796) |
| | | *Dardanus calidus* (Risso, 1827) |
| | | *Diogenes pugilator* (P. Roux, 1829) |
| | Paguridae | *Cestopagurus timidus* (P. Roux, 1830) |
| | | *Pagurus anachoretus* Risso, 1827 |
| | | *Pagurus chevreuxi* (Bouvier, 1896) |
| | | *Pagurus cuanensis* Bell, 1845 |
| | | *Pagurus prideaux* Leach, 1815 |
| Brachyura | Dromiidae | *Dromia personata* (Linnaeus, 1758) |
| | Homolidae | *Homola barbata* (Fabricius, 1793) |
| | Ethusidae | *Ethusa mascarone* (Herbst, 1785) |
| | Eriphiidae | *Eriphia verrucosa* (Forskål, 1775) |
| | Leucosiidae | *Ilia nucleus* (Linnaeus, 1758) |
| | Progeryonidae | *Paragalene longicrura* (Nardo, 1869) |
| | Epialtidae | *Acanthonyx lunulatus* (Risso, 1816) |
| | | *Herbstia condyliata* (Fabricius, 1787) |
| | | *Herbstia nitida* Manning & Holthuis, 1981 |
| | | *Lissa chiragra* (Fabricius, 1775) |
| | | *Pisa armata* (Latreille, 1803) |
| | | *Pisa nodipes* Leach, 1815 |
| | Inachidae | *Achaeus cranchii* Leach, 1817 |
| | | *Inachus dorsettensis* (Pennant, 1777) |
| | | *Macropodia czernjawskii* (A.T. Brandt, 1880) |
| | | *Macropodia rostrata* (Linnaeus, 1761) |
| | Majidae | *Eurynome aspera* (Pennant, 1777) |
| | | *Maja crispata* Risso, 1827 |
| | | *Maja squinado* (Herbst, 1788) |
| | Pilumnidae | *Pilumnus hirtellus* (Linnaeus, 1761) |
| | | *Pilumnus minutus* De Haan, 1835 |
| | | *Pilumnus spinifer* H. Milne Edwards, 1834 |
| | Portunidae | *Carupa tenuipes* Dana, 1852 |
| | | *Charybdis hellerii* (A. Milne-Edwards, 1867) |
| | | *Portunus hastatus* (Linnaeus, 1767) |
| | Xanthidae | *Xantho pilipes* A. Milne-Edwards, 1867 |
| | Grapsidae | *Pachygrapsus marmoratus* (Fabricius, 1787) |
| | Percnidae | *Percnon gibbesi* (H. Milne Edwards, 1853) |
| | Pinnotheridae | *Nepinnotheres pinnotheres* (Linnaeus, 1758) |

The species most frequently reported from Mediterranean marine caves is *Stenopus spinosus*, found in 41 caves (out of 133: 30.8%), followed by *Palaemon serratus* (34 caves: 25.6%) and *Herbstia condyliata* (32 caves: 24.1%). Six other species were reported from more than ten caves and a further six species from at least seven caves (Figure 2). All the remaining species were found in six (<5%) caves or less, with 30 species having been found in a single cave only. Thus, the majority of the 76 species reported from Mediterranean marine caves should apparently be considered as accidental in such habitats with the exception of *S. sketi*, which is hitherto known only from the cave where it has been

described [33], and *O. addaia*, known only from two marine caves: one in Menorca (Spain) [32] and one in Marseilles (France) [34].

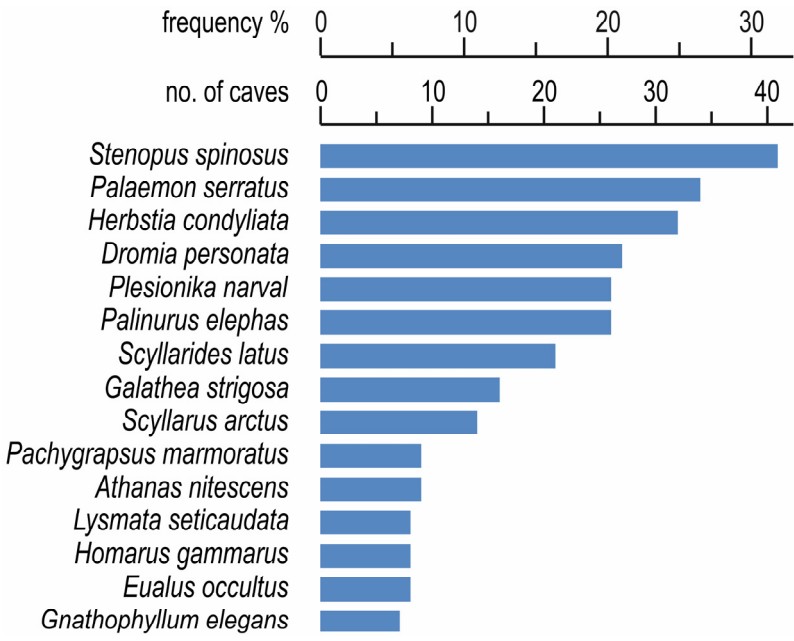

**Figure 2.** The 15 most common decapod species in Mediterranean Sea caves. Species found in less than seven caves (5%) are not considered.

## 3. Regional Variations in Species Richness

The 76 decapod species are not distributed homogenously among the 133 caves. The most species-rich cave is the Grotte de l'Île Plane (Gulf of Lion), with 19 species, followed by the Grotta Marina di Bergeggi (Ligurian Sea), with 15 species. The Grotta della Cala di Mitigliano (South Tyrrhenian Sea) and the Grotte del Ciolo (Ionian Sea) host 13 species each, while 12 species have been reported from the Grotte de la Triperie (Gulf of Lion) and the Spilja Vrbnik (Adriatic Sea), and 11 species from the Túnel Llarg (Balearic Sea) and the Fará Cave (Aegean Sea). All the remaining caves have less than ten species each, 59 of them having only one species.

Marine caves are highly fragmented habitats, whose connectivity is expectedly low [35]. In addition, the Mediterranean Sea is compartmentalized into fairly isolated sub-basins, which display a great variety of climatic and hydrologic conditions. Bianchi et al. [36] recognized a number of geographic sectors within the Mediterranean Sea, different for oceanographic characteristics and biota composition. Information on cave decapods is available for 10 sectors (Figure 1): (1) the Alboran Sea, immediately east of the Straits of Gibraltar; (2) the Balearic Sea and the Sea of Sardinia, from eastern Spain to western Sardinia; (3) the Gulf of Lion, from Cap de Creus (Catalonia) to the Giens Peninsula (France); (4) the Ligurian Sea, from the Giens Peninsula to the Piombino Promontory (Italy); (5) the North Tyrrhenian Sea, from the Piombino Promontory to the Pontine Islands (Italy); (6) the South Tyrrhenian Sea, from the Pontine Islands to north Sicily; (7) the Adriatic Sea, between Italy and the Balkan Peninsula; (8) the Ionian Sea, between Italy and western Greece; (9) the Aegean Sea, between Greece and Turkey; (10) the Levantine Sea, from south Turkey to Israel.

This choice of sectors patently suffers from the heterogeneity of research effort (in terms of number of caves explored), which has been maximum in the northern Mediterranean and minimum in southern and extreme western areas. Perhaps not coincidentally, the greatest number of decapod species has been recorded in the sectors of the Gulf of Lion and Ligurian Sea. Clearly, the number of species found is positively correlated to the number of caves explored [14], which varied from a minimum of three,

in the Alboran Sea, to a maximum of 22, in the Ionian Sea, being comprised between seven and 20 in the remaining sectors (Table 2).

**Table 2.** Summary of the main parameters and coefficients of the species/caves curves.

| Sectors | $N$ | $S_N$ | $c$ | $z$ | $r_{dir}$ | $S_\infty$ | $r_{rec}$ |
|---|---|---|---|---|---|---|---|
| Alboran Sea | 3 | 5 | 1.7 | 1.00 | 1.000 | 32 | 0.993 |
| Balearic and Sardinia seas | 10 | 18 | 5.1 | 0.58 | 0.989 | 27 | 0.999 |
| Gulf of Lion | 7 | 29 | 7.5 | 0.74 | 0.996 | 45 | 0.999 |
| Ligurian Sea | 13 | 24 | 3.8 | 0.74 | 0.996 | 48 | 1.000 |
| North Tyrrhenian Sea | 16 | 19 | 3.1 | 0.69 | 0.991 | 32 | 0.999 |
| South Tyrrhenian Sea | 12 | 20 | 3.3 | 0.74 | 0.998 | 37 | 0.999 |
| Adriatic Sea | 20 | 19 | 2.1 | 0.75 | 0.996 | 32 | 0.999 |
| Ionian Sea | 22 | 22 | 2.3 | 0.64 | 0.997 | 36 | 0.999 |
| Aegean Sea | 18 | 17 | 3.1 | 0.60 | 0.990 | 23 | 0.999 |
| Levantine Basin | 9 | 15 | 2.8 | 0.78 | 0.997 | 37 | 1.000 |

$N$: number of caves surveyed in the different sectors; $S_N$: cumulative number of species in $N$ caves; $c$ and $z$: coefficients calculated from the relation $S = c \cdot N^z$; $r_{dir}$: correlation coefficient of the curve resulting from the direct plot of the cumulative number of species against the number of caves (see Figure 3); $S_\infty$: theoretical maximum number of species in each sector, by extrapolation; $r_{rec}$: correlation coefficient between the reciprocals of both the cumulative number of species and the number of caves.

Thus, to make comparable the cave decapod species richness in the different sectors of the Mediterranean Sea, the cumulative number of species has been plotted against the number of caves surveyed in each sector (Figure 3). Curves were fitted to these plots according to the equation $S = c \cdot N^z$, where $S$ is the number of species, $N$ is the number of caves, $c$ gives the number of species that may be expected in one cave and $z$ is the slope of the regression line relating $S$ and $N$ [37,38]. Plotting the reciprocals of the cumulative number of species against the reciprocals of the number of caves was also tried. This allowed an extrapolation to show the hypothetical number of species to be found with an 'infinite' number of caves, i.e., the theoretical maximum number of cave species in each sector [39]: this number of species is useful for inter-regional comparisons but must not be taken for a real prediction.

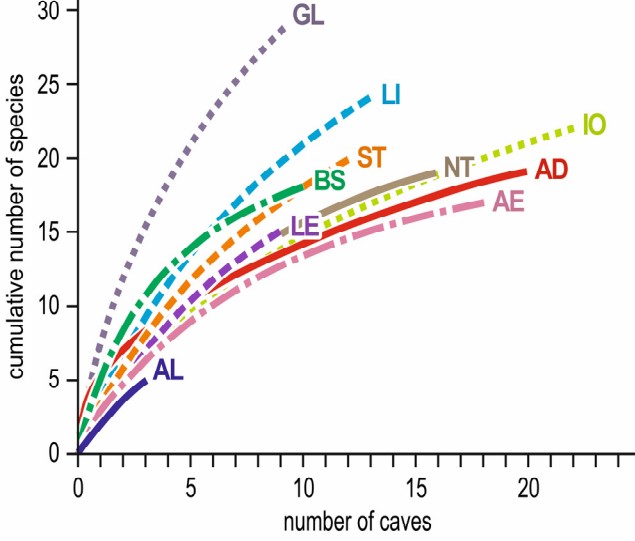

**Figure 3.** Species accumulation curves for cave decapods in ten Mediterranean Sea sectors. AL = Alboran Sea; BS = Balearic and Sardinia seas; GL = Gulf of Lion; LI = Ligurian Sea; NT = North

Tyrrhenian Sea; ST = South Tyrrhenian Sea; AD = Adriatic Sea; IO = Ionian Sea; AE = Aegean Sea; LE = Levantine Sea (see Figure 1 for sectors location).

The equation $S = c \cdot N^z$ fitted well to all ten species/caves curves (Table 2). The curve for the Gulf of Lion lays above all the remaining curves (Figure 3) due to high values of both $c$ and $z$. When $z$ exceeds 0.5, species richness may be sufficiently explained by environmental heterogeneity alone [40]: as $z$ values were distinctly higher than 0.5 for all sectors, it can be inferred that differences among individual caves are of major importance for increasing the regional cave species pool.

Similarly, the plots of the reciprocals of species against cave numbers were well fitted by regression lines of the form $1/S = a \cdot 1/N + b$: the reciprocal of the intercept $b$ represents the number of species $S_\infty$ to be found in an infinite (i.e., 1/0) number of caves. The highest value of $S_\infty$ was found in the Gulf of Lion, followed by the Ligurian Sea. In all, the Mediterranean sectors richest in cave decapod species are the most northern ones, independent of research effort (number of caves explored, the actual number of surveys within each individual cave being not available).

## 4. Zoogeography

Most species have a distributional range restricted to a geographical area, and species having similar ranges can be grouped in chorological categories. The fauna of the Mediterranean Sea is composed of species belonging to several chorological categories [41]. These categories are not uniformly distributed through the whole basin but tend to occur more or less abundantly in the different geographic sectors of that sea [36,42]. The most 'typical' Mediterranean fauna, rich in endemics [43], occurs in the central sectors and especially in the western basin [44]. The northern and colder sectors harbour boreal Atlantic species (ice-ages remnants, especially of the Würm glacial), while the southern and warmer sectors are colonized by (sub)tropical species [45]. The Alboran Sea, located immediately east of Gibraltar, exhibits stronger Atlantic affinities thanks to the continued penetration of Atlantic fauna with the incoming flux of water [46]. On the contrary, the Levantine Sea is experiencing an important influx of Red Sea species since the opening of the Suez Canal [47]—a phenomenon known as 'Lessepsian immigration' in recognition of Ferdinand de Lesseps, the French diplomat who promoted the cut of the canal [48].

The present decapod dataset provides an opportunity to test whether Mediterranean marine caves exhibit the same chorological composition as the geographic sectors to which they belong or represent a filter that favours some chorological categories with respect to others. Mediterranean marine caves are considered as refuges for archaic fauna, often of Tethyan or Pliocene origin, which escaped the competitive pressure by more modern species [6,49]. Thus, marine caves would be expected to host a higher proportion of endemic and tropical species (both Tethyan and Pliocene fauna being typically tropical in character) and little or no recent immigrants in comparison to other shelf habitats.

Mediterranean cave decapod species have been assigned to six chorological categories, based on the review of d'Udekem d'Acoz [50] integrated with subsequent studies [27,51–54], to compute the chorological spectra of the Mediterranean as a whole and of the ten geographical sectors outlined above (Figure 4): (1) Mediterranean endemics; (2) Atlantic–Mediterranean species; (3) Lusitanian(–Boreal) species; (4) Mauritanian(–west African) species; (5) circum(sub)tropical species; (6) non-indigenous species (NIS), including both aliens and cryptogenics.

The endemics include six cave decapod species. Three are long known to live exclusively in the Mediterranean Sea (see Table S21 by C. Froglia in [55]): *Hippolyte holthuisi*, which has recently been recognized as the Mediterranean vicariant of the Atlantic congeneric *H. varians* Leach, 1814 [56], *Maja squinado* and *Periclimenes amethysteus*. *Odontozona addaia* and *Salmoneus sketi* are hitherto known only from the Mediterranean Sea [32–34]. Finally, *Caridion* sp. was originally identified with the boreal species *C. steveni*

Lebour, 1930 by Ledoyer [57], who found it in a cave at Villefranche-sur-Mer (France), but considered as a distinct, still undescribed species by subsequent authors [50,58].

**Figure 4.** Chorological spectra of cave decapods in the Mediterranean Sea as a whole and in ten marine sectors. NIS = non-indigenous species.

Atlantic–Mediterranean species typically thrive in the East Atlantic from the English Channel (La Manche) to the North, to Morocco and Mauritania to the South, including Macaronesia (Azores, Madeira and Canary Islands) and the Mediterranean Sea [59,60]. Five decapod species found in Mediterranean caves strictly belong to this category: *Eriphia verrucosa*, *Pachygrapsus marmoratus*, *Palinurus elephas*, *Pisidia bluteli*—a senior synonym of *P. longimana* (Risso, 1816) and distinct from the similar *P. longicornis* (Linnaeus, 1767) according to Ferreira and Tavares [61]—and *Scyllarus arctus*. Thirteen other species in this category have a wider distribution in the East Atlantic, reaching up to Norway to the North and Cape Verde Islands or even Namibia to the South: *Achaeus cranchii*, *Athanas nitescens*, *Diogenes pugilator*, *Dromia personata*, *Eualus cranchii*, *E. occultus*, *Eurynome aspera*, *Galathea intermedia*, *Pagurus cuanensis*, *P. prideaux*, *Palaemon elegans*, *Pilumnus hirtellus* and *Xantho pilipes*.

Lusitanian species are here defined as those Mediterranean–Atlantic species that are restricted to the Mediterranean and the western European coasts from Portugal to Brittany and do not extend southward [59]. The term Lusitanian is here adopted in its classical acceptation and according to its etymology (Lusitania being the name that Romans used to indicate a region that roughly corresponds to the present-day Portugal); other authors used the term in a stricter or wider sense [62–64]. Lusitanian decapods reported from Mediterranean caves include 11 species: *Cestopagurus timidus*, *Clibanarius erythropus*, *Eualus sollaudi*, *Lissa chiragra*, *Lysmata seticaudata*, *Macropodia czernjawskii*, *Munida rugosa*, *Nepinnotheres pinnotheres*, *Pagurus chevreuxi*, *Pisa nodipes* and *Synalpheus gambarelloides*. Nine other species stretch further north (up to Norway), thus displaying a boreal

character: *Galathea dispersa*, *G. nexa*, *G. strigosa*, *Homarus gammarus*, *Inachus dorsettensis*, *Macropodia rostrata*, *Palaemon adspersus*, *P. serratus* and *Porcellana platycheles*.

Similarly, Mauritanian species are those whose range, outside the Mediterranean Sea, reaches Mauritania or even Cape Verde Islands but does not extend northward [59,65]. Cave decapods belonging to this category include 13 species: *Dardanus calidus*, *Ethusa mascarone*, *Gnathophyllum elegans*, *Ilia nucleus*, *Lysmata nilita*, *Maja crispata*, *Pagurus anachoretus*, *Palaemon xiphias*, *Paragalene longicrura*, *Periclimenes scriptus*, *Pilumnus spinifer*, *Scyllarides latus* and *Scyllarus pygmaeus*. Nine other species extend their range further southward along the coasts of west Africa: *Acanthonyx lunulatus*, *Alpheus dentipes*, *Balssia gasti*, *Calcinus tubularis*, *Herbstia condyliata*, *Homola barbata*, *Pisa armata*, *Portunus hastatus* and *Stenopus spinosus*.

Three Mediterranean cave decapod species have a circum(sub)tropical range, thriving in all warm waters of the world ocean: *Brachycarpus biunguiculatus*, *Dardanus arrosor* and *Plesionika narval*.

Finally, Mediterranean caves host seven non-indigenous species (NIS) of decapods with different origin: *Percnon gibbesi* is an amphiamerican and amphiatlantic (sub)tropical species rapidly spreading in the Mediterranean Sea thanks to the passive drift of larvae with currents [66]; *Herbstia nitida* comes from tropical west Africa [30]; *Carupa tenuipes*, *Charybdis hellerii*, *Pilumnus minutus* and *Saron marmoratus* are Indo-Pacific species that possibly penetrated into the Mediterranean Sea through the Suez Canal [67]; *Urocaridella pulchella* represents an example of a species first described from the Mediterranean Sea, where it is obviously an alien of Indo-Pacific origin [68], and only later found in its native range [69]. As a whole, all these alien species share a clear tropical affinity.

Analysing the chorological spectra of the decapod fauna in Mediterranean marine caves (Figure 4), the proportion of NIS results is 9%, i.e., less than half the figure for the whole Mediterranean [55], suggesting that marine caves, and especially internal cave portions, are comparatively less receptive to newcomers [70]. The highest proportion of alien decapods is found in the eastern sectors, as expected [71]. The proportion of endemics in caves is 7.9%, i.e., slightly less than the corresponding figure of 9.9% for the total decapod fauna of the Mediterranean [55]. Endemics are comparatively more represented in the north-western sectors than in south-eastern sectors, conforming to a common pattern for the East Mediterranean marine fauna [36,72]. On the contrary, the Aegean Sea and the Levantine Basin contain more circum(sub)tropical species with respect to the northern sectors. Such a picture, however, is probably undergoing change as sea water warming is allowing warm-water species to establish also in the northern reaches of the Mediterranean Sea [73]: a still unpublished—and, therefore, not included in the present dataset—record of *B. biunguiculatus* comes from a marine cave near Marseilles (https://doris.ffessm.fr/Forum/Crevette-des-grottes-38429, accessed on 23 February 2022). Differences among geographic sectors are apparent also for Atlantic–Mediterranean, Lusitanian(–Boreal) and Mauritanian(–west African) species and partly mirror those of the total decapod fauna of the Mediterranean. These general patterns, although apparently reliable, should be taken with caution: more research is needed, especially in those geographic sectors where cave decapod fauna has not been sufficiently investigated (see, for instance, the Alboran Sea, with only five species reported from three caves).

## 5. Depth and Cave Zone Preferences

Information about the water depth of the seafloor at which the cave opens was available for 89 out of the 133 (66.9%) Mediterranean marine caves stored in the decapod dataset, representing nine countries (Albania, Croatia, Cyprus, France, Greece, Israel, Italy, Spain and Turkey) and ten geographic sectors (Adriatic Sea, Aegean Sea, Alboran Sea, Balearic and Sardinia seas, Gulf of Lion, Ionian Sea, Levantine Basin, Ligurian Sea, North Tyrrhenian Sea and South Tyrrhenian Sea). The water depth at the base of cave entrances ranges from 1 m to 52 m. In these 89 caves, 59 decapod species have been

recorded, but only the 14 species found in at least five caves have been considered for statistical analysis.

Most of these 14 species have a wide depth range (Figure 5), but *Palaemon serratus* and *Lysmata seticaudata* clearly prefer shallow waters, never extending deeper than 20 m. *P. serratus* is commonly found outside caves in infralittoral habitats [74]. *Scyllarides latus, Herbstia condyliata* and *Dromia personata* also prefer shallow depths but were recorded down to 30 m. *Plesionika narval* and *Homarus gammarus*, on the contrary, avoid the shallowest caves and were more frequently recorded in deep caves, down to 52 m and 45 m, respectively. Outside caves, *P. narval* is mostly found at epibathyal depths [75]. All the remaining species prefer intermediate depths and typically show a circalittoral affinity outside caves [76].

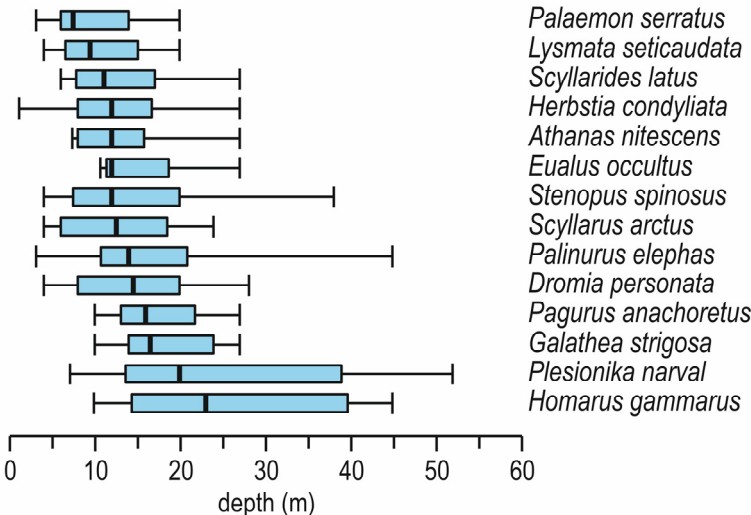

**Figure 5.** Depth preference of selected cave decapod species (depth is that of the seafloor at the entrance of the cavity). Boxes depict the 25–75 percent quartiles; the thick vertical line inside the box indicates the median; the whiskers represent minimal and maximal values.

Independent of the depth at which they open, marine caves are topographically and ecologically heterogeneous in their inside, exhibiting dramatic environmental gradients [77]: within a few metres, there are variations of light, water movement and trophic input, which, in the external environment, can take place within tens or even hundreds of metres [11]. These environmental gradients generate a marked zonation of cave communities [78]. Riedl [7] distinguished six biotic zones, based on species replacement across the outside–inside gradient of blind-ended caves. Bianchi and Morri [79] also distinguished six ecological zones, but, rather than species replacement, they considered change in growth forms, trophic guilds, three-dimensional structure and biotic cover of the sessile communities. Due to the great influence of the French school on Mediterranean marine ecologists, the most widely accepted and followed model of cave zonation is that of Pérès and Picard [44], who distinguished two basic situations based on the occurrence of characteristic species: the semidark cave, and the dark cave.

For the purpose of the present review, it has been possible to distinguish three cave zones where decapods have been recorded: the entrance, the semi-dark zone and the dark zone. Information about which zone decapods came from is available from 39 caves (29.3%) situated in all the ten geographical sectors (Adriatic Sea, Aegean Sea, Alboran Sea, Balearic and Sardinia seas, Gulf of Lion, Ionian Sea, Levantine Basin, Ligurian Sea, North Tyrrhenian Sea and South Tyrrhenian Sea) and belonging to nine countries (Albania, Croatia, France, Greece, Italy, Lebanon, Malta, Spain and Turkey). In these 39 caves, 60 decapod species have been found. However, only the twelve species recorded in more than five caves have been considered to make possible an estimation of their relative frequency in the three cave zones (Figure 6).

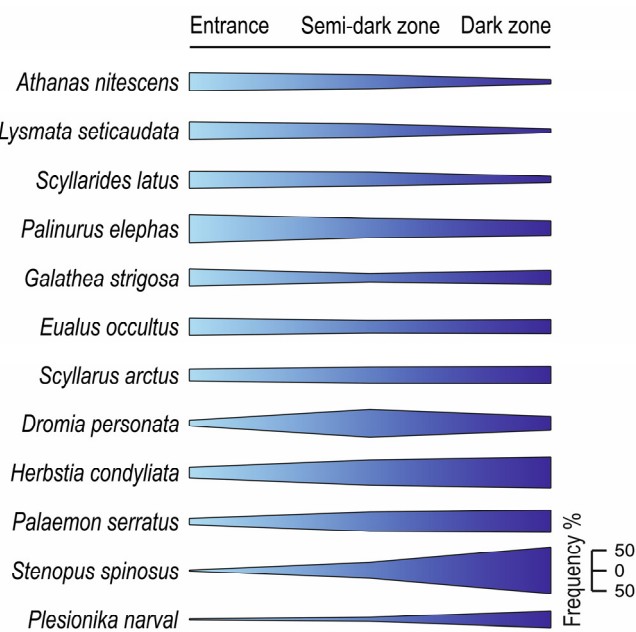

**Figure 6.** Kite diagrams illustrating cave zone preference for selected cave decapod species.

Three species, namely *Athanas nitescens*, *Lysmata seticaudata* and *Scyllarides latus*, have been more frequently reported from the entrance of the caves. On the contrary, *Plesionika narval* and *Stenopus spinosus* were more frequent in the dark zone. For the remaining seven species (*Dromia personata*, *Eualus occultus*, *Galathea strigosa*, *Herbstia condyliata*, *Palaemon serratus*, *Palinurus elephas* and *Scyllarus arctus*), no clear pattern is discernible. Pessani and Manconi [26] analysed decapod zonation in the Grotta della Cala di Mitigliano (South Tyrrhenian Sea, Italy): out of the 14 species found, only *H. condyliata* reached the innermost part of the cave, while the remaining species were localized near the entrance and in the semidark zone. Harmelin [80] ascribed *Homarus gammarus*, *L. seticaudata*, *P. serratus*, *P. elephas*, *S. latus* and *S. arctus* to the semidark zone; *G. strigosa*, *H. condyliata* and *S. spinosus* to the dark zone.

## 6. Ecology and Distribution in the Grotta Marina di Bergeggi

The distribution of decapods within Mediterranean marine caves has been described by Gili and Macpherson [25] in Spain and by Pessani and Manconi [26] in Italy. However, no study has addressed the relation between decapod species occurrence within a marine cave and environmental data. Sgorbini et al. [19] took measurements in situ and collected water and sediment samples for laboratory analysis in order to define the environmental characteristics of the Grotta Marina di Bergeggi, a comparatively small but morphologically complex marine cave in the Ligurian Sea, NW Italy (Figure 7).

Besides a wide emerged part, the Grotta Marina di Bergeggi has a submerged part that develops between the sea surface and 7 m depth, has a length of about 40 m and is articulated in five main topographic features. The Remo's Cavern is the outer cave, with an environmental situation comparable to the adjacent sea tract, but a reduced illumination. The Gulley, the two lateral chambers (First Chamber and Lights' Chamber) and the Hall are obscure and lie along the overall tunnel path of the cavity. The inner 'lakes' (Lemons' Lake and Lake through the Hole) represent blind-ended portions, obscure and subject to meteoric water infiltrations. The parameters taken into account included light (μW·cm$^{-2}$), current velocity (cm·sec$^{-1}$), water temperature (°C), salinity (psu), dissolved oxygen (ppm), pH, sediment mean grain size (mm), suspended organic matter (SOM, mg·l$^{-1}$), chlorophyll *a* (μg·l$^{-1}$) and nitrogen to carbon (N/C) percent ratio (Table 3). The data were taken in July 1986; the methodological details about measurement and sampling can be found in Sgorbini et al. [19].

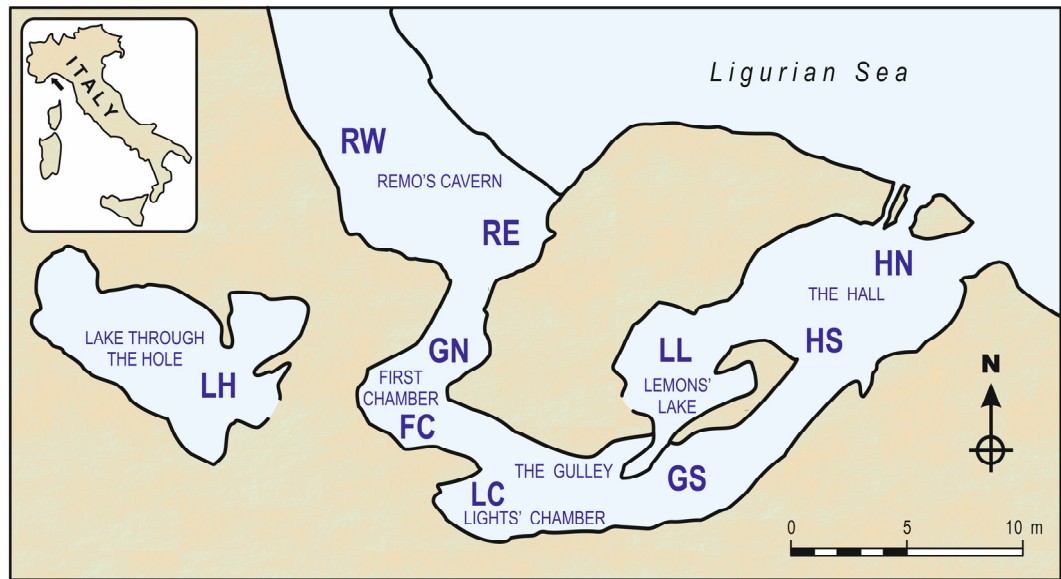

**Figure 7.** Simplified map of the submerged part of the Grotta Marina of Bergeggi with main toponyms and sampling points. RW = Remo's Cavern West; RE = Remo's Cavern East; GN = Gulley North; GS = Gulley South; FC = First Chamber; LC = Lights' Chamber; HN = Hall North; HS = Hall South; LL = Lemons' Lake; LH = Lake through the Hole. Inset: location of Bergeggi in Italy.

**Table 3.** Environmental data measured in July 1986 in different stations of the Grotta Marina di Bergeggi [19,20].

| Parameters | RW | RE | GN | GS | FC | LC | HN | HS | LL | LH |
|---|---|---|---|---|---|---|---|---|---|---|
| Light ($\mu$W·cm$^{-2}$) | 29.0 | 32.5 | 0.4 | 0.1 | 0.2 | 0.0 | 1.3 | 0.2 | 0.1 | 0.0 |
| Current Velocity (cm·s$^{-1}$) | 20.3 | 15.9 | 14.3 | 14.1 | 7.5 | 1.6 | 15.9 | 14.3 | 7.6 | 2.7 |
| Temperature (°C) | 19.0 | 19.0 | 18.8 | 18.5 | 19.0 | 18.5 | 18.3 | 18.0 | 17.5 | 17.0 |
| Salinity (psu) | 37.8 | 37.8 | 37.7 | 37.5 | 37.1 | 37.5 | 37.6 | 37.2 | 26.8 | 27.8 |
| $O_2$ (ppm) | 6.25 | 5.55 | 6.49 | 6.13 | 6.35 | 6.10 | 5.35 | 5.80 | 6.45 | 3.53 |
| pH | 8.22 | 8.18 | 8.15 | 8.10 | 8.11 | 8.09 | 8.14 | 8.14 | 7.82 | 7.90 |
| Mean Grain Size (mm) | 7.5 | 10.0 | 6.9 | 7.0 | 7.9 | 4.8 | 9.0 | 7.5 | 0.5 | 3.6 |
| Suspended Organic Matter (mg·l$^{-1}$) | 1.05 | 1.05 | 0.79 | 0.39 | 0.79 | 0.25 | 1.74 | 1.96 | 0.73 | 0.39 |
| Chlorophyll *a* ($\mu$g·l$^{-1}$) | 0.30 | 0.30 | 0.30 | 0.25 | 0.30 | 0.20 | 0.30 | 0.26 | 0.14 | 0.15 |
| Nitrogen/Carbon % | 11.2 | 11.8 | 10.3 | 10.0 | 9.8 | 9.3 | 8.7 | 8.1 | 9.8 | 9.4 |

RW = Remo's Cavern West; RE = Remo's Cavern East; GN = Gulley North; GS = Gulley South; FC = First Chamber; LC = Lights' Chamber; HN = Hall North; HS = Hall South; LL = Lemons' Lake; LH = Lake through the Hole.

Knowing the value of these parameters within the cave allows evaluating the relative importance of three main ecological factors: illumination, hydrological confinement and trophic depletion [13]. The decrease in light has been considered the most obvious factor influencing species distribution, through the limitation of the development of photophilic assemblages [81]. The threshold values of light intensity in the different cave zones have rarely been measured: Southward et al. [82] showed that, at the cave entrance, with light intensity equal to 17% of that of the surface, the assemblages are still photophilic; at 3%, the assemblages become sciaphilic, below 0.8%, only animals are found. According to Harmelin et al. [6], light intensity in the dark zone is lower than 0.01% of the sea-surface value. Early observations by Harmelin [83] showed that tunnel-shaped dark caves exhibited a richer biota than blind-ended dark caves. This led to the idea that, besides light, hydrological confinement was a major driver of fauna distribution in marine caves [84]. Hydrological confinement is a complex and rather abstract quantity: it is mainly a

hydrodynamic notion, essentially depending on water exchange, a factor that has rarely been measured. Bianchi et al. [85] measured current velocities ranging from less than 2 cm·s⁻¹ to 6–10 cm·s⁻¹, but sea conditions may greatly alter these figures. Water circulation affects a series of hydrological parameters, such as temperature, salinity, oxygen concentration, pH and sedimentation. The absence of vegetal life, and, hence, of autochthonous primary production, makes marine cave communities completely dependent on the input of food from the external environment [77]. Therefore, animals living in the innermost portions of the cave, far from the entrance, suffer trophic depletion [86] in terms of both the quantitative decrease of the nourishment (e.g., reduced amount of suspended organic matter and chlorophyll *a* content) and its qualitative degradation (e.g., low nitrogen to carbon ratio). The amount of suspended particulate matter decreases significantly from the entrance to the innermost portions of the marine caves due to the progressive sedimentation of the suspended particles and their capture by filter-feeders [87]. Chlorophyll *a* characterizes 'fresh' vegetal organic matter, rich in live phytoplankton cells, and its concentration in the water decreases dramatically in the innermost parts of the cave [88]. The nitrogen to carbon ratio is an index of the nutritional value of the organic matter: food with high carbon and little nitrogen content (e.g., cellulose) is poorly nutritious compared with food that is proportionally richer in nitrogen (e.g., proteins). Inside caves, the ratio can get lower than 6%, a threshold value for animal consumption [89,90].

Cluster analyses on all these environmental parameters in the Grotta Marina di Bergeggi allowed recognizing five locales that correspond well to the topographic features of the cavity [20]. In two stations for each locale (Figure 7), the presence of decapod species was recorded visually by scuba diving in July 1986 during daytime. Their abundance was estimated semi-quantitatively in the field using the following codes [91]: 1 = one individual; 2 = two to five individuals; 3 = more than five individuals.

In total, 15 species were found: *Herbstia condyliata* (Figure 8a), *Stenopus spinosus* (Figure 8b), *Palaemon serratus* (Figure 8c), *Lysmata seticaudata* (Figure 8d) and *Scyllarus arctus* (Figure 8e), in the order, were the most abundant and common; *Dromia personata* (Figure 8f), *Palinurus elephas* and *Alpheus macrocheles* were scarcer, while *Clibanarius erythropus* (Figure 8g), *Eriphia spinifrons*, *Inachus dorsettensis* (Figure 8h), *Pachygrapsus marmoratus*, *Pilumnus hirtellus*, *Scyllarides latus* (Figure 8i) and *Xantho pilipes* were occasional (only one individual found). The highest number of species was found in the Hall, followed by the Gulley; the lowest number was found in the lakes (Figure 9).

A correspondence analysis was applied to the data matrix of 15 decapod species × 10 stations to explore the ecological gradients [92]. Groups of species were individuated by cluster analysis using Euclidean distances and minimum variance clustering [93]. All the statistical analyses were performed using the free software PaSt [94].

The cluster analysis distinguished four groups of species (Figure 10). The first group contains four species (*P. marmoratus*, *A. macrocheles*, *E. spinifrons* and *P. hirtellus*) that were restricted to the Remo's Cavern. The second group includes seven species (*S. latus*, *X. pilipes*, *P. elephas*, *L. seticaudata*, *C. erythropus*, *S. arctus* and *I. dorsettensis*) that preferred the parts of the cave nearest to the entrances, such as the First Chamber and the northern stations of the Gulley and the Hall. The third group is composed by three species (*H. condyliata*, *D. personata* and *S. spinosus*) that, on the contrary, preferred the innermost parts of the cave but not the lakes: the southern stations of the Hall and Gulley and the Light's Chamber. Finally, the fourth group is made by *P. serratus*, the only species that—despite being typically infralittoral—preferred the lakes.

An interesting property of correspondence analysis is the possibility of plotting both station-points and species-points on the same factorial plane, which allows for an immediate reading of gradients and affinities. Only the first axis from the correspondence analysis was significant ($p < 0.05$) according to the Lebart's test [95]. Both species-points and station-points of the Grotta Marina di Bergeggi ordered along the first axis according to the external–internal gradient (Figure 10).

**Figure 8.** Underwater photographs of decapod species in the Grotta Marina of Bergeggi: (**a**) *Herbstia condyliata*; (**b**) *Stenopus spinosus*; (**c**) *Palaemon serratus*; (**d**) *Lysmata seticaudata*; (**e**) *Scyllarus arctus*; (**f**) *Dromia personata*; (**g**) *Clibanarius erythropus* (in a shell of *Cerithium vulgatum*); (**h**) *Inachus dorsettensis*; (**i**) *Scyllarides latus*.

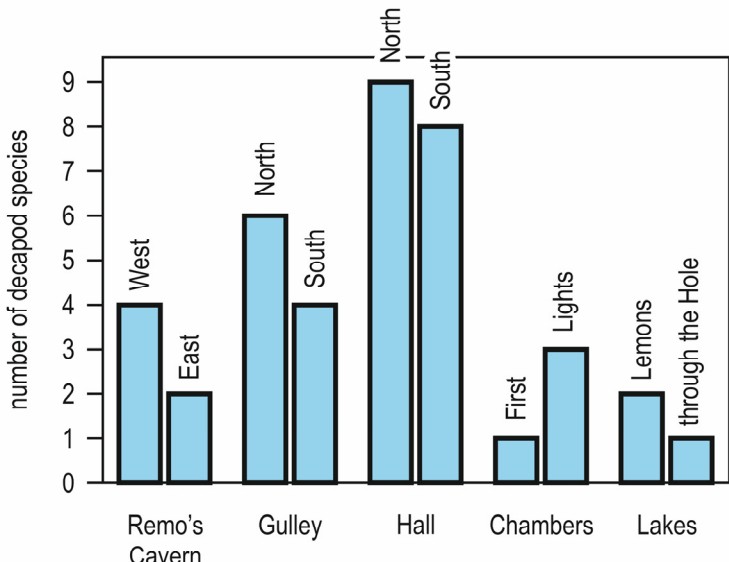

**Figure 9.** Number of decapod species in the different locales and stations of the Grotta Marina of Bergeggi.

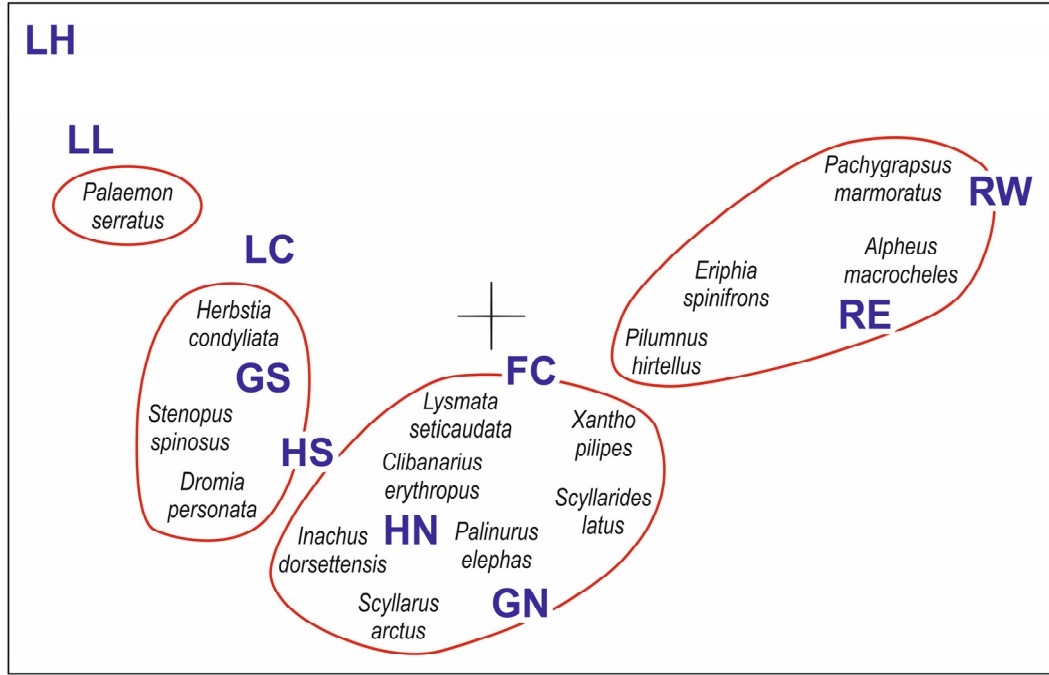

**Figure 10.** Bivariate plot on the plane of the first two axes from correspondence analysis of decapod species and stations in the Grotta Marina of Bergeggi. First (horizontal) axis explains 46.1% of the total variance; second (vertical) axis explains 25.4% of the total variance. Only first axis is significant ($p < 0.05$). Species are grouped according to cluster analysis (Euclidean distance, minimum variance clustering). LH = Lake through the Hole; LL = Lemons' Lake; LC = Lights' Chamber; GS = Gulley South; HS = Hall South; HN = Hall North; GN = Gulley North; FC = First Chamber; RE = Remo's Cavern East; RW = Remo's Cavern West.

To explore which of the environmental parameters measured is most likely responsible for that gradient, a correlation analysis was performed between the station-point scores of the first axis and the values of the individual parameters in such stations (Figure 11). The first axis scores were highly significantly correlated ($p < 0.01$) with light intensity, and significantly correlated ($p < 0.05$) with the nitrogen to carbon ratio. No significant correlation was found between the first axis scores and the remaining parameters illustrative of hydrological confinement (current velocity, temperature, salinity, oxygen, pH and mean grain size) or trophic depletion (suspended organic matter and chlorophyll *a*). This result suggests that light is the main factor influencing the ecological distribution of decapods in marine caves. Apart from some of the species occasionally found at the cave entrance, as in the Remo's Cavern, decapods entering caves are sciaphilic species that find shelter during the day and get out at night to feed in the external environment [76]. Motility allows decapods to avoid the negative effects of hydrological confinement and trophic depletion, which strongly influence the abundance and distribution of the sessile fauna [13].

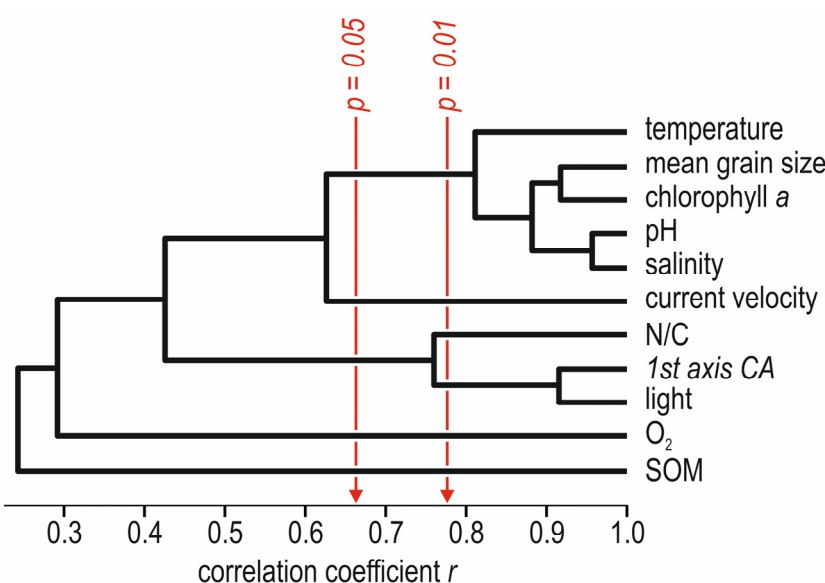

**Figure 11.** Dendrogram (minimum variance clustering) of the correlation coefficient *r* between the scores of the first axis from correspondence analysis, applied to the matrix of 15 decapod species × 10 stations in the Grotta Marina di Bergeggi, and a set of environmental parameters measured within the cave in the same stations where decapod species were recorded. *p* is the significance level.

### 7. Final Remarks

Thanks to the exploitation of a large dataset originally developed by Gerovasileiou and Voultsiadou [14], integrated with new information purposely assembled, the present review unveiled for the first time the decapod species richness of Mediterranean marine caves. The total of 76 species reported from 133 caves in 13 countries corresponds to a momentous increase with respect to previous regional or partial inventories.

The greatest number of cave decapod species has been recorded in the northernmost sectors of the Mediterranean, such as the Gulf of Lion and the Ligurian Sea, which is only in part due to the higher number of caves explored [14]. A general decrease in species richness from the north-western to south-eastern sectors is a common pattern for the whole Mediterranean Sea biota [55]. The proportion of endemic species in marine caves is comparatively low, suggesting that the role of refuge that marine caves exert for the survival of relict species known for other taxa [6,49] is not confirmed in the case of decapods. Consistently, marine caves, although less receptive than other habitats, are nonetheless subjected to the colonization by alien decapods, as anticipated for other faunal groups [70]; the presumed Lessepsian species *Urocaridella pulchella*, first reported from a marine cave in Turkey [96], has rapidly spread to marine caves of Cyprus [97] and Greece [98].

Most decapod species were found only in a few marine caves and can, therefore, be considered as 'accidental', having entered caves by chance. Many of them are simply sciaphilic species widespread in infra- or circalittoral habitats, which enter marine caves in search of shelter. The case of the Grotta Marina di Bergeggi demonstrated that decapod occurrence in marine caves is mainly correlated to the decrease in light intensity and not to other factors (hydrological confinement, trophic depletion) that characterize the marine cave environment [13].

The fact that the bulk of the decapod fauna in marine caves is made by accidental species, drawn by the regional species pool, implies that the chorological spectra of decapods found in marine caves mostly reflect that of the geographical sectors where the caves are located. Warm-water species are more represented in the south-eastern sectors, and so are non-indigenous decapods, which typically exhibit tropical affinities. Mediterranean endemics are more represented in the western sectors (Balearic and Sardinia seas, Gulf of Lion), conforming to the general pattern for the Mediterranean fauna [36].

Only 15 decapod species were comparatively frequent (more than 5% of the caves), but none of them seem exclusive to caves and have, therefore, to be defined as 'stygoxenes' (occur in caves but do not complete their life cycle within caves) or at most 'stygophiles' (can complete their life cycle within caves but also thrive in suitable habitats outside caves). Unfortunately, little is known about the population biology of cave decapods. Seasonal observations through one year allowed Denitto et al. [29] to detect large swarms of juveniles of *Palaemon* sp. (identified as *P. elegans* by the authors; however, considering the cave zones where those shrimps were collected, we do believe that, in reality, they belonged to *P. serratus*, whose juveniles have a shorter rostrum and can be misidentified with *P. elegans*) in spring and especially in summer in the innermost and dark portion of the Grotta di Ciolo (SE Italy), which suggests recruitment within the cave. In the Grotta Marina di Bergeggi (NW Italy), juveniles of *Herbstia condyliata* were observed in October (CNB, unpublished observations). All decapods produce planktonic larvae, but whether species living in caves release their larvae within or outside the cave is not known; however, juveniles have been seen in caves, and zoeae have been collected in cave plankton [99].

Other putative stygophiles include *Stenopus spinosus* and *Plesionika narval*, which combined frequent occurrence in caves and preference for the dark zone. The concept of 'secondary stygobiosis' has been conceived for those crevicular (cryptobiotic) or bathyphilic species that only secondarily colonize caves and become abundant there [100]. While commonly reported from marine caves, *S. spinosus* can be found also in rock crevices and biogenic reef anfractuosities and inside shipwrecks [101]. *P. narval* is actively fished with bottom trawls or pots in epibathyal grounds 200 to 400 m deep [75] but its swarms can be easily observed in marine caves 20 to 50 m deep [102]: consistently, the analysis of a large Mediterranean dataset for the present review highlighted that it prefers deep caves.

Morphological and behavioural adaptations to cave life typical of stygobiotic species can be observed in caridean and stenopodid shrimps as well as in galatheid and brachyuran crabs found in marine caves [103]. Two species have not been—at least yet—reported from habitats other than marine caves and thus could perhaps be considered as 'stygobionts' (i.e., cave-exclusive): the stenopodid *Odontozona addaia*, from the Balearic Sea and the Gulf of Lion [34], and the alpheid *Salmoneus sketi*, from the Adriatic Sea [33]. Future research in cryptic and deep-sea habitats might substantiate or refute their presumed stygobiosis.

The existing knowledge on Mediterranean marine cave decapods is far from being complete: only a small number of existing caves have been explored and, mostly, in a superficial and incomplete manner, while many caves (especially the deepest ones) are still unknown. Future research should focus on filling regional gaps (e.g., south-eastern Mediterranean, Alboran Sea) and is expected to lead to an increment in the number of species known and to help answer many fundamental questions in cave biology and ecology [104]. An aspect of cave decapod biology that deserves further investigation is their role in cave ecosystem functioning. As other motile cave species do, decapods likely exit the caves to feed in nearby external habitats at night and return to rest in caves during daytime. During their stay in the cave, they release faecal pellets, which increase the internal trophic load [13, and references therein]. This way, they import organic matter, thus mitigating trophic depletion in caves [105]. Such a pivotal role has been thoroughly investigated in the swarm-forming mysid *Hemimysis speluncola* [106] and in the schooling cardinal fish *Apogon imberbis* [107] but not in decapods. In addition, large decapod crustaceans, such as *Dromia personata*, provide an opportunity for the transport (phoresy) of sessile filter-feeding organisms, which settle as epibionts on their carapace [86]. These epibionts are allowed feeding outside at night and represent a source of larvae capable of maintaining pseudo-populations of sessile stygoxenic taxa in caves, thus contributing to their biodiversity [13].

Quantitative data on cave decapod populations in different periods of the year are virtually lacking [29] but are badly needed to tackle population biology topics, which are of major importance to understand behavioural adaptations and for conservation purposes. Mediterranean marine caves are a priority habitat, protected by the Habitats Directive of the European Union and by the Mediterranean Action Plan of the United Nations Environment Programme, but nonetheless threatened by many anthropogenic pressures [18,87,108,109]. The lists of Mediterranean marine invertebrates that are protected, according to Appendix III of the Bern Convention (https://www.coe.int/en/web/bern-convention, accessed on 23 February 2022), or whose exploitation is regulated, according to Annex III of the Barcelona Convention (https://ec.europa.eu/environment/marine/international-cooperation/regional-sea-conventions/barcelona-convention/index_en.htm, accessed on 23 February 2022), include six species of decapods that occur inside marine caves: *Homarus gammarus*, *Maja squinado*, *Palinurus elephas*, *Scyllarides latus*, *Scyllarus arctus* and *S. pigmaeus*. For *H. gammarus* and all Palinuridae (hence including *P. elephas*), European Council Regulation No 1967/2006 defines the minimum catchable size, prohibits capturing and selling berried females and imposes that the specimens caught accidentally be promptly released back to the sea. Basic ecological interest and the need for conservation initiatives combine to claim for intensifying the investigations on the decapod fauna of the Mediterranean Sea caves.

**Author Contributions:** Conceptualization, C.N.B. and C.M.; methodology, C.N.B., C.F., V.G. and C.M.; software, V.G.; validation, C.F.; formal analysis, C.N.B. and C.M.; investigation, C.N.B., V.G. and C.M.; resources, C.N.B., C.F., V.G. and C.M.; data curation, C.N.B. and V.G.; writing—original draft preparation, C.N.B., C.F., V.G. and C.M.; writing—review and editing, C.N.B., C.F., V.G. and C.M.; visualization, C.N.B. and C.M.; supervision, C.F.; project administration, C.M.; funding acquisition, C.N.B. and C.M. All authors have read and agreed to the published version of the manuscript.

**Funding:** This review received no external funding.

**Institutional Review Board Statement:** Not applicable.

**Data Availability Statement:** Data are available from the authors upon request: V.G. for the Mediterranean Sea marine caves database, C.N.B. for the Bergeggi cave dataset.

**Acknowledgments:** We thank Valerio Zupo (Ischia, Italy) for stimulating an earlier version of our review, and Bella Galil (Haifa, Israel) for advice on *Urocaridella pulchella*. Giovanni Diviacco (Genoa, Italy) led the earliest marine biology studies on the Grotta Marina di Bergeggi. C.N.B. and C.M. wish to dedicate this paper to their unforgettable colleague and friend Riccardo Cattaneo-Vietti (1949–2021), who coined the concept of secondary stygobiosis, here applied to crustacean decapods for the first time.

**Conflicts of Interest:** The authors declare no conflict of interest.

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
