# Peer review of "Distribution and Ecology of Decapod Crustaceans in Mediterranean Marine Caves: A Review"

_diversity, doi:10.3390/d14030176_

Round 1

Reviewer 1 Report

General comment: the AA might deal extensively with the the increase of organic matter in the cave, due to decapods presence (faeces….).

Reviewer 2 Report

The proposed review together with included original results represent a respectable, important contribution to the knowledge and ecological relationships of decapods inMediterranean caves. The readers profit from rich and exhaustive information.

Minor note: 

Line 178: " ... independently of research effort (number of caves explored)". In combination with Fig. 3 this statement insinuates that all studied caves contained some decapods. If not so, inform the reader about total numbers documented. Note, that the curves in Fig. 3 are sensitive to zero counts (if any).

Reviewer 3 Report

General remarks:

Bianchi et al. have reviewed existing knowledge of the ecology of decapod crustaceans occurring in Mediterranean marine caves.

I found the article to be well-written and comprehensive, not only summarising existing information but drawing new insights on the ecology of decapods by pooling together the data obtained from various published sources. Although two of the authors (VG and CNB) have recently published a review synthesizing current knowledge of Mediterranean marine caves (Oceanography and Marine Biology: An Annual Review, 2021, 59, 1-88), I do not consider this an issue as the extent of overlap between that review and the present article is rather minimal.

Specific comments:

[1] It is a real pity that the authors have not considered adding supplementary material. The article would benefit greatly from having a Supplementary File that gives the following information for each of the 133 caves considered: (i) Name; (ii) Geographical coordinates; (iii) List of decapod species reported in the cave; (iv) List of literature sources providing the information on decapods in that cave.

[2] Several figures have low resolution, and will need to be replaced by better quality ones. Most of the figure and table legends are a bit too brief and should be more informative, allowing readers to understand the content of the figures/tables without needing to refer to the main text.

[3] Table 1 should be elaborated to indicate, for each species, whether it can be considered as a stygobiont, stygophile, stygoxene or simply an accidental occurrence. If such a detailed classification is not possible, at least a simple categorisation into “known mainly from caves” vs “known also from outside caves” would still be helpful to include. Some comments along these lines are included in the draft manuscript but only towards the end as part of the ‘Final remarks’.

[4] In the section on species richness, the authors refer to the parameter ‘z’ as a measure of species richness, but this is commonly considered to be a measure of species accumulation rather than richness per se. They also conclude that the “sectors richest in cave decapod species are the most northern ones, independently of research effort (number of caves explored).” However, this conclusion is ignoring the fact that the review also reported most decapods are only ‘accidentally’ found in caves; thus, the research effort is not determined solely by the number of caves surveyed but more importantly by the total number of surveys per sector, given that caves surveyed multiple times will likely have a greater decapod richness than those surveyed only once. The analysis should preferably be repeated taking this into account. If not, than this limitation ought to be acknowledged.

[5] There seems to be something wrong with the scale of the x-axis of Figure 3, and the total number of caves surveyed for the different sectors as given in this figure does not always agree with that reported in Table 2. For example, in the case of Sector AE, 19 caves were surveyed according to Figure 3, but according to Table 2 this should be 18 caves.

[6] It is not clear how Figure 6 was constructed. The legend does not give sufficient details. The main text mentions the “relative frequency” of the species in the different cave zones, but there is no indication of how this was estimated. For example, based on the scale included in the figure, it seems that the values for P. narval recorded in the three zones would not sum up to 100%, and contrarily, those of S. spinosus add up to more than 100%. How can this be?

[7] The authors should check again the manuscript for any inconsistencies. For example, the statement in Lines 268-269 that “marine caves, and especially internal cave portions, are comparatively little receptive to newcomers” is contradicted by the one in Line 490 that “marine caves turned out quite receptive to alien decapod species”.

[8] I believe that the article would benefit from several other minor revisions, which I have outlined for the authors in the attached annotated version of the manuscript.
